# Understanding In-context Learning with a Pelican Soup Hypothesis

## Abstract

Motivated by Pelican Soup riddles, we propose a hypothesis, the *Pelican Soup Hypothesis*, to explain the in-context learning ability of large language models. We propose a simple but general formalism for natural language classification problems. With this formalism, we show how to understand in-context learning as the generalization of modeling some linguistic phenomena under distribution shifts. We provide evidence supporting this hypothesis. First, we synthesize a dataset called *Calcutec* that replicates the linguistic phenomena and show that language models trained with this dataset acquire in-context learning ability and benefit from chain-of-thought. Second, our experiment of GPT-2 on some natural language tasks shows the linkage between one of the linguistic phenomena and in-context learning. Third, we use a digit addition task to inspect one of the identified distribution shift type and find that larger models generalize better. Our contributions offer a way to better understand how and why in-context learning works, and our Calcutec and digit addition tasks will facilitate future studies on in-context learning.

## 1 Introduction

Large language models (Brown et al., 2020; Chowdhery et al., 2022) have demonstrated the ability to perform downstream natural language processing (NLP) classification tasks via in-context learning. That is, they can perform a task based on some demonstrations, i.e. a few input-label pairs in the context they condition on (Brown et al., 2020). Wei et al. (2022a) described this as an ability that emerges as we scale up the models. Moreover, providing reasoning steps (e.g., chain-of-thoughts) in the context can further boost the performance (Wei et al., 2022b; Nye et al., 2021). Because language models are trained with general text (e.g., from the web) instead of text sequences in prompts' format, it is mysterious why language models can perform in-context learning. It is also unclear why scaling up the model size and providing reasoning is helpful. Following previous studies on language characteristics that lead to the in-context learning ability (Xie et al., 2022; Hahn & Goyal, 2023; Chan et al., 2022), we propose a *Pelican Soup Hypothesis*, aiming to provide a new perspective to understand the above phenomena.

We propose a hypothesis, the *Pelican Soup Hypothesis*, which is inspired by Xie et al. (2022) and a lateral thinking puzzle game, Pelican Soup. A Pelican Soup game involves a puzzle master who has a story in their mind. The goal of the participants is to recover the story by asking the puzzle master yes/no questions. An observation is that, once the participants recover the story, they will be able to answer any questions about the story. Therefore, the story has a similar role as the latent variable in Xie et al. (2022)'s framework which defines the input-output mapping, and the yes/no questions are similar to the demonstrations for in-context learning.

Given the above observation, we can study in-context learning by considering why humans can solve Pelican Soup riddles. We conjecture that this is because the person who makes the story and the ones who solve the riddle share the same (or similar) commonsense (McCarthy, 1960) about logical relationships among things in this world (Schank & Abelson, 1988). Similarly, the definition and interpretation of any NLP task's instruction also relies on the commonality of commonsense shared among people. We formalize this in § 2.

The formalism elucidates the connections between modeling text and in-context learning. First, general text often contain step-by-step logical reasoning steps. Modeling them allows language

model to acquire a logical model of the world and thus is helpful for solving downstream tasks (§ 3.1). Second, modeling coreferences and name entities in general text is similar to solving Pelican Soup riddles (§ 3.2). Therefore, we can view in-context learning as the language modeling task under distribution shifts, such as the absence of reasoning steps (§ 4).

Based on the above observation, we propose our Pelican Soup hypothesis: in-context learning in language models can be explained as generalization under several types of distribution shifts. We provide three pieces of empirical evidence to support this hypothesis. In § 5, we construct a synthetic dataset—Calcutec—and show that models trained on this dataset exhibits in-context learning ability. We also observe the efficacy of chain-of-thoughts with our synthetic dataset. In § 6, we conduct an experiment with natural language and show that using pronouns as verbalizers can lead to comparable performance (with GPT-2 Large), which is aligned with our hypothesis that modeling coreferences contributes to in-context learning. Finally, in § 7, we study the distribution shift caused by the absence of reasoning steps with a digit addition task. We find that larger models are better at gaining an intuition that allows them to jump to a conclusion without reasoning steps. This may help explain the emergence of in-context learning ability when we scale up the model size.

To summarize, our main contributions are:

- In §2, we propose a general formalism for NLP classification tasks, which may facilitate future NLP theory research.

- We propose the Pelican Soup hypothesis, which explains in-context learning based on linguistic phenomena we find pervasive in general text and the generalization under distribution shifts.

- We propose a synthetic dataset, Calcutec. With it, we can train GPT-2-sized models and observe phenomena that large language models have, including the emergence of in-context learning and the efficacy of chain-of-thought. It can serve as a test bed for studies on in-context learning and model architecture.

- Through the digital addition task we propose, we show that larger models can more effectively gain intuition. Future studies may utilize this task to study the mechanism of the emergent capability.

## 2 A FORMALISM FOR NLP CLASSIFICATION TASKS

We posit that a knowledge base is indispensable for solving any NLP task as cognitive psychologists have suggested that semantics of words are based on the interrelation among concepts (Fodor, 1975; 2008; Siskind, 1996; Murphy, 2004). For example, the semantic of the verb "move" is embodied in its implication on "position". A knowledge base that allows a system to utilize the semantics of the word "move" may contain a rule: if X moved, then the position of X changed. Therefore, we follow the early formulation of AI (McCarthy, 1960; Croft, 1993) to include a "commonsense" knowledge base KB in our formalism of NLP tasks.

Seeing that people share a commonsense knowledge base KB, we propose a simple formalism: For any objective or prescriptive NLP classification task (Rottger et al., 2022) that classifies an input $x$ to one of $|\mathcal{Y}|$ classes, it can be described with some instructions $t$ and some descriptions $z_i$ for $i = 1, 2, \cdots, |\mathcal{Y}|$ such that $x$ is labeled as $y_i$ if and only if

$$\text{KB} \wedge t \wedge x \models z_i. \tag{1}$$

That is, $x$ belongs to class $y_i$ if and only if based on the commonsense rules in KB and the instruction $t$, $x$ entails $z_i$. Note that because we can rewrite Eq. 1 as $\text{KB} \wedge x \models t \rightarrow z_i$, we can combine the task instructions to the description of each class $z_i$ for $i = 1, 2 \cdots, |\mathcal{Y}|$. Therefore, we can represent any NLP classification tasks as $\langle z_1, z_2, \cdots, z_{|\mathcal{Y}|} \rangle$.

For example, we can formulate the sentiment analysis task over movie reviews as $\langle z_+, z_- \rangle = \langle$ "I like the movie", "I dislike the movie"$\rangle$. We all have the commonsense in our KB that people would only recommend something they like. Thus, based on this rule, we can derive the label of an input "I would recommend this movie."

How well this formalism can describe NLP classification tasks depends on to what extent we can map a natural language to a formal language. There has been a long history of linguistics studying

the relationships between natural languages and formal languages (Carnap et al., 1968; Bresnan & Bresnan, 1982; Steedman, 1987; 1996; Sag et al., 1999), such as first-order logic (Frege et al., 1879; Peirce, 1883). The expressive power of this formulation also depends on the formal language used. Note that propositional logic is enough to simulate any Turing machines (Cook, 1971; Levin, 1973). We postulate that Turing machine is powerful enough to solve reasonable NLP classification problems. Therefore, this formulation is general enough for most of the NLP classification tasks.

## 3 CONNECTION BETWEEN LANGUAGE MODELING AND IN-CONTEXT LEARNING ABILITY

In this section, we qualify the characteristics of language that may contribute to language models' in-context learning ability. We formalize these characteristics in §5.1 with concrete assumptions.

### 3.1 LANGUAGE MODELING LEARNS REASONING WITH THE KNOWLEDGE BASE

Since people write articles based on similar KBs of commonsense, language models may be able to acquire the KB by modeling general text. Additionally, language models may learn to do reasoning with the rules in the KB, because articles generally contain statements that are logical and coherent and proceed like a proof induction process.

Such kind of articles may be pervasive in the training data. Essays arguing some claims are one example. To be convincing, these essays should contain statements following the rules in the KB and proceed like a proving process that induces their conclusions. Articles describing a series of events can be another example. Although the series of events may be surprising, in general, they still follow commonsense and at least do not disobey physics rules in the world. Therefore, by modeling these articles, a language model can not only learn the commonsense rules in KB, but also learn to utilize these rules for induction.

### 3.2 LANGUAGE MODELING LEARNS LATENT CONCEPT INFERENCE

Based on the formulation in § 2, a skill required to solve a Pelican Soup riddle or perform in-context learning is to recover $z_y$ for all $y \in \mathcal{Y}$ based on the input-label pairs. We argue that similar skills are also required to model natural language on many occasions.

Modeling coreferences is one such scenario. A characteristic of natural language is that we can associate a latent situation with a pronoun or a name entity such as "Alice" or "Bob". Such association can be very arbitrary as long as it is consistent throughout an article. Therefore, sometimes language models are able to predict the correct pronoun or entity as the next token by recovering the latent situation they refer to. For example, in the context "He told me he didn't see her in the class, so I think the bad news may be about", in order to know that the continuation is "her" instead of "him", the model needs to figure out that "she" may be a person to whom something unexpected happened. This process is similar to recovering $z_y$ for class $y$. Also, the model can only make this conjecture based on the mention of "him" and "her" in the context, which has a similar role as the sample-label pairs in the demonstrations for doing in-context learning. Therefore, modeling general text is similar to performing in-context learning. This may explain why large language models are able to perform in-context learning with labels associated to irrelevant verbalizers (Wei et al., 2023) and also provide an alternative explanation for the linkage between in-context learning and emergent abilities found by Lu et al. (2023).

We may see commonly used subjective verbalizers such as "good", "bad" as special cases of pronouns. Unlike pronouns, these subjective verbalizers are not associated with latent situations as arbitrarily as pronouns. In general, "good" and "bad" are associated with some desirable and undesirable qualities respectively. However, there is still some freedom. Things that are good in one sense may be bad in another sense. Different authors also have different senses of value. Additionally, how "good" and "bad" are associated with latent concepts should be consistent throughout an article. Therefore, the language modeling objective is still helpful for the model to learn to do in-context learning with these verbalizers.

# 4 Distribution Shifts from Language Modeling to In-context Learning

In § 3, we show that performing in-context learning is similar to modeling some linguistic phenomena in general text. In this section, we describe the distribution shifts from the general text to the prompts for in-context learning.

**Absence of Reasoning Steps.** In § 2, we formulate that an input $x$ belongs to class $y$ if $\mathrm{KB}, x \models z_y$, meaning that we can induce $z_y$ from KB and $x$ with a few induction steps. Induction processes like this may be included in the training data, as we argue in § 3.1. However, when performing in-context learning, the reasoning steps are absent in the input. Therefore, it is different from the general text used to train the model.

**Structure Mismatch.** We can see the sequence of input-label pairs in the demonstration as short paragraphs about the same entities relevant to $z_y$ represented by a verbalizer $y$. This is similar to a general article where paragraphs are around a few entities. However, in a demonstration, the verbalizer representing the subject, $y \in \mathcal{Y}$, is always at the end of a input-label pair. This is different from general text, where entities may be mentioned at any place in the paragraph.

**Verbalizer Mismatch.** In § 3.2, we argue that the pronouns can be the main source where language models learn the in-context learning ability. However, people do not use pronouns as verbalizers in general (Brown et al., 2020; Min et al., 2022a;b). Those commonly used verbalizers do not appear in the training data as frequently as pronouns either.

# 5 Experimenting with Synthetic Datasets

Based on the arguments above, we propose our toy setting, *Calcutec*, consisting of data generation processes for general text and downstream tasks. We will use the generated datasets to study whether language models trained with the general text can generalize well under the distribution shifts described in § 4. This will provide evidence supporting our Pelican Soup hypothesis.

## 5.1 Assumptions

We list the assumptions based on which we construct our toy setting, Calcutec. Theses assumptions formalize the intuitions described in §3.

**Assumption 1** (Setup)**.** *A sentence in natural languages can be mapped to a logic formula in a logic model (Tarski, 1956), e.g. propositional logic. The logic model has a set of variables $\Sigma$ that serves as the set of atom concepts (as aligned with cognitive psychology theories, e.g. Fodor (1975; 2008); Piantadosi (2021)). Additionally, there is $\mathrm{KB}$, which is a set of formulas in the logic model. Datasets in this language are generated based on the rules defined in this $\mathrm{KB}$*

**Assumption 2** (Paragraph in an Article)**.** *A paragraph in a general article represents a proving process with some induction steps dropped. Specifically, given some premises and a goal formula, we can write down a proving tree that induces to the goal formula by applying induction rules in the logic model. A paragraph is a traversal of the nodes of the proving tree in the topological ordering. Because in the real world, we may not write down our reasoning process exactly step-by-step, we assume that some of the nodes are randomly dropped.*

**Assumption 3** (Article)**.** *Let $S$ be a set of formulas representing a topic (which can be a few entities). An article about $S$ consists of a set of paragraphs in each of which a formula in $S$ is present.*

**Assumption 4** (Pronoun)**.** *There is a set of symbols $\Gamma$ simulating the pronouns in articles. $r \in \Gamma$ can be associated to a set of formulas $Z_r$. This association is arbitrary but coherent throughout each article. Namely, given an article where $r$ is associated with $Z_r$, for any formula $z \in Z_r$, $z$ present in the article, can be replaced by $r$. We will also use symbols in $\Gamma$ as the verbalizers.*

**Assumption 5** (Downstream Task)**.** *NLP classification tasks can be defined with the formulation in § 2. This implies that a task can be defined by the descriptions $\{z_y\}_{y \in \mathcal{Y}}$.*

## 5.2 CALCUTEC

Based on the above assumptions, we propose our toy dataset, Calcutec. Sentences in these datasets are formulas in a logic model. The logic model has a set of variables $\Sigma$. Additionally, there is a knowledge base KB consisting of a set of formulas. We generate articles for language model training and input-label pairs for downstream tasks based on the formulas in KB.

**Setup.**  Following Assumption 1, we construct a pseudo-language with the following components:

- Logic model: We use a subset of propositional logic as our logic model. We only consider Horn clauses (Horn, 1951), formulas in the form $A \land B \to C$. This allows us to efficiently generate proof with time complexity linear to the size of the knowledge base (Dowling & Gallier, 1984).
- Atom concepts: We use 100 variables as our set of atom concepts $\Sigma$.
- KB: For each variable $\sigma \in \Sigma$, there are 5 formulas in the knowledge base whose consequence (the literals after $\to$ in a horn clause) is $\sigma$, while the antecedents (the literals before $\to$) are sampled from $\Sigma \backslash \{\sigma\}$ uniformly at random. Overall, the knowledge base has 500 formulas.
- Following Assumption 4, we have a set $\Gamma = \{r_i\}_{i=1}^4$ representing the pronouns. .

**Training Dataset.**  Following Assumption 3, an article is a concatenation of paragraphs about a topic $S$ separated by ";". In our synthetic language model training dataset, each article contains 16 paragraphs. We define the topic of each article as 2 randomly sampled variables, say $S = \{s_1, s_2\} \subset \Sigma$. We generate a paragraph based on Assumption 2 in the following step:

1. We pick a variable $s$ from $S$ uniformly at random.
2. We randomly generate a proof for KB, $P \models g$, where $P \subset \Sigma$ is the premise and $g \in \Sigma$ is the goal of the proof. We ensure that this proof contains the topic $s$.
3. We convert the proof tree to a sequence of proving steps by traversing the proving tree in a topological order with ties broken randomly. Each node in the proof tree corresponds to a rule in KB, so the resulting sequence of proving steps consists of horn clauses in the form $a_1 a_2 \to b$. We separate the clauses in the sequence with commas.
4. We rewrite the first step of the proving process to contain the premises of the proof. [1]

Each article is the concatenation of paragraphs separated with a semicolon and ends with a period. To simulate the fact that articles in the real world always skip some reasoning steps, we further apply some perturbations over the generated paragraphs that drop some reasoning steps with a skip rate $p_{skip}$ (details in Appendix A). [2]

After we generate an article, we randomly draw two pronoun symbols $r_a$, $r_b$ from $\Gamma$. We use $r_a$ to represent the topic $S$, replacing $s \in S$ in the article with $r_a$. We then find the two most frequent symbols in the article and replace them with $r_b$. (We provide the pseudo code Alg. 1 in Appendix.)

**Downstream Tasks.**  Following the formulation in Assumption 5, we define a binary classification task by defining the description $z_+$ and $z_-$ for the positive and negative classes respectively. $z_+$ and $z_-$ are the disjunctions of atom concepts, i.e. in the form of $a_1 \lor a_2 \lor \cdots$. We create several downstream tasks using several different disjunctions. Each input is a subset of variables in $\Sigma$. When generating the inputs, we ensure that only one of the classes can be induced from an input. We elaborate more about how we generate the inputs in the next section.

**Demonstration.**  We represent an input-label pair as $x_1 x_2 \cdots \to r$, where $x_1 x_2 \cdots$ is the input part and $r \in \{r_+, r_-\} \subset \Gamma$ is a pronoun that serves as a verbalizer.

**Chain-of-thought.**  A chain-of-thought is in the format same format as the training data, but ends with a pronoun symbol $r \in \{r_+, r_-\}$, e.g. $x_1 x_2 \cdots \to x_3; x_3 \cdots x_4 \to r_+$. This chain-of-thought reflects the step-by-step induction process from the inputs to the label.

---

[1]We find that this step is necessary to prevent the language model trained on it from hallucinating irrelevant variables randomly. It is important for our experiment for chain-of-thought, but is not necessary for language models to learn the in-context learning ability.

[2]Models can acquire in-context learning ability even with $p_{skip} = 0$ (Figure 7 in Appendix).

Table 1: Calcutec examples for training, in-context learning (ICL), and chain-of-thought (CoT).

| Train | x57 x56 x64 r3 → x79 , r1 x57 → x58 , x90 x58 → r3 , r3 r1 → x20 , ... ; x80 x66 x63 x83 x1 → x82 , x80 x82 → r1 , ... , x64 x80 → x54 . |
|---|---|
| ICL | x44 x67 x34 x62 → r2 ; x55 x38 x50 x48 → r1 ; x21 x59 x57 x86 → r2 ; x55 x76 x84 x99 → |
| CoT | x44 x67 x34 x62 → x16 , x34 x62 → x99 , x99 x16 → r1 ; x77 x34 x62 x97 → x12 ... ; x21 x59 x57 x86 → x69 , x59 x57 → x75 , x69 x75 → r2 ; x55 x76 x84 x99 → |

## 5.3 INSPECTING THE DISTRIBUTION SHIFTS

Our Calcutec dataset allows us to inspect some types of distribution shifts in § 4. Calcutec replicate the absence of reasoning steps in the in-context learning setting. It also replicates the structure mismatch distributional shift because the target symbols $R$ rarely appear at the end in our training data. Additionally, we make a few design choices to inspect other distribution shifts:

**Verbalizer Mismatch** When we are picking the pronoun symbols in $R$, we assign probability mass 45%, 45%, 5%, 5% to $r_1, r_2, r_3, r_4$. In this way, we can inspect whether using less frequent verbalizers will lead to worse performance.

**Unseen Tasks** We investigate whether the model can generalize to a new combination of concepts that is unseen in the training data. Therefore, when we are generating our training data, we ensure that the pronouns are never associated with some combinations of atom concepts (e.g. the topic $s_1, s_2 \in \Sigma$). We then test the trained model on tasks where $z_+$ and $z_-$ are the disjunctions of unseen combinations. Additionally, while in the training data, each pronoun is associated with only 2 atom concepts, we also test whether the model can perform in-context learning on tasks where $z_+$ and $z_-$ are the disjunctions of three atom concepts.

(a) The proof tree a paragraph in the training dataset corresponds .

(b) A balanced tree for a downstream task sample.

Figure 1: Proof trees examples.

**Unseen Inference Process** Based on Assumption 2 and Assumption 5, an input-label pair of a downstream task is similar to the premise-pronoun pair in a paragraph. To examine whether the trained model can generalize well when the induction process for the label is different from the induction process for the pronoun in the training data, we generate a training dataset where all the pronouns are induced from the premise with a left-branching proof tree with a depth equal to 2 (Figure 1a), while the test data contains samples whose labels are induced from the input with balanced trees (Figure 1b).

## 5.4 EXPERIMENT DETAILS

We use $p_{skip} = 0.25$ in our experiment. We generate 60,000 articles with 16 paragraphs following the process described in § 5. Among them, we use 10k for validation. We train a 6-layer Transformer (Vaswani et al., 2017) model until the loss on the validation set converges.

## 5.5 EXPERIMENT RESULTS AND DISCUSSION

Figure 2 shows that the model trained with Calcutec can perform in-context learning, indicating that the model can generalize under the distribution shifts described in §4. This evidence supports our Pelican Soup hypothesis. We further inspect the in-context learning performance under the distribution shifts described in §5.3:

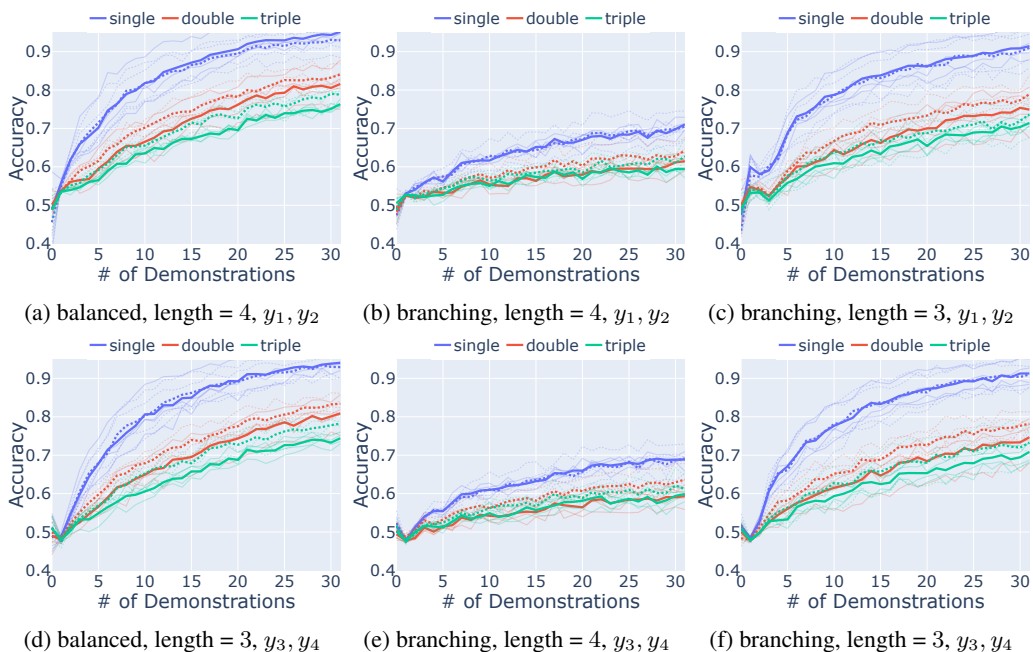

Figure 2: In-context learning accuracy with Calcutec when using different verbalizers ($y_1, y_2$ or $y_3, y_4$) and input lengths (3 or 4). The dotted lines represent the performance of *unseen combinations* described in §5.3, while the different colors represent the number of atom concepts in $\Sigma$ each class ($z_+$ or $z_-$) is associated to. The main lines represent the average accuracy of 5 tasks. We plot the performance of each task in lighter colors.

- Infrequent verbalizer: We observe similar performance of using frequent pronouns ($r_1, r_2$) or infrequent pronouns ($r_3, r_4$) as the verbalizers.

- Unseen tasks: Figure 2 shows that the model has similar performance over tasks defined with unseen combinations of atom concepts (dot lines) as over tasks defined with seen combinations (solid lines). The models can also generalize to tasks defined with three concepts (green lines).

- Unseen Inference Process: When the depth of the proving tree required to solve the task is the same, the tree structure does not affect the performance much. In the branching, input length = 4 setting, the proof tree has depth 3, which is greater than the depth of $r \in \Gamma$ in the training data. We observe that the performance in this setting is worse, yet still nontrivial.

In sum, the results show that the model can generalize well under several distributional shifts.

We experiment with 4-shot learning using chain-of-thought. The results in Table 2 show that the model also benefits from chain-of-thought. Interestingly, using chain-of-thought mitigates the accuracy gap between the balance and branching setups when input length is 4. We conjecture that it is because chain-of-thought makes it easier to uncover the concepts the verbalizers are associated with.

Table 2: The 4-shot accuracy of in-context learning (ICL) versus chain-of-thought (CoT) when using $r_1, r_2$ as verbalizers. Using $r_3, r_4$ leads to similar results (Table 5).

| Task | Balanced | | Branching | |
|---|---|---|---|---|
| | ICL | CoT | ICL | CoT |
| Single | 68.5 | 89.8 | 57.1 | 91.7 |
| Double | 58.5 | 76.1 | 53.5 | 76.3 |
| Triple | 57.0 | 68.2 | 53.0 | 73.0 |

# 6 REAL-WORLD EVIDENCE

As evidence of out Pelican Soup Hypothesis, we show that even a small language model can do in-context learning with pronouns as the verbalizers. We use *"[input]", [verbalizer] thought.* as a task-agnostic template and "he", "she" as label verbalizers. We follow the setup in Min et al. (2022a) and compare the accuracy of the binary classification tasks using GPT2-Large. Even though

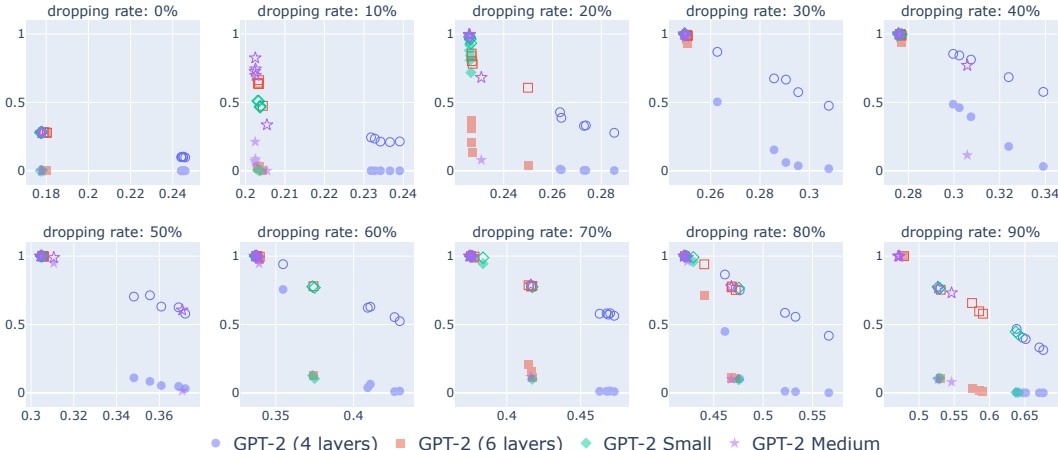

Figure 3: The exact accuracy (y-axis, solid points) and digit-level accuracy (y-axis, hollow points) versus validation loss (x-axis) for the 5-digit addition task for 5 random seeds and different dropping rates $p_{drop}$ (We include the results of the 6-digit addition task Figure 10 in the appendix.)

Wei et al. (2023) shows that only larger models can do in-context learning with task-irrelevant verbalizers, our results indicate that this task-agnostic template with pronouns is competitive with those task-specific templates when using a small model. It may indicate that modeling pronouns contribute to the in-context ability of language models.

## 7 DIGIT ADDITION TASK

To overcome the distribution shifts caused by the absence of reasoning steps (§ 4), a model needs to be able to generate the final answer without generating the intermediate reasoning steps. We refer to this ability as *intuition*. Since in the Calcutec environment, it is difficult to construct a task that requires long reasoning process without exponentially increasing the size of $\Sigma$, we use another toy task, the digit addition task, to inspect to what extreme the model can acquire the intuition for long reasoning processes.

Table 3: The accuracy of using task-specific templates/verbalizers (direct) (Min et al., 2022a) v.s. using task-agnostic templates/pronouns for 16-shot in-context learning with GPT2-Large.

| task | direct | pronoun |
|------|--------|---------|
| SST-2 | 63.0 | 65.3 |
| CR | 61.7 | 62.9 |
| MR | 59.2 | 56.7 |
| Subj | 51.0 | 62.2 |

### 7.1 SETUP

The digit addition task requires the model to perform the addition operation over two $n$-digit numbers. It requires $n$ steps to solve if we solve it digit-by-digit. We generate a training set consisting of digit-by-digit reasoning steps for 100,000 pairs of numbers. Each of the intermediate steps may be dropped at a constant probability $p_{drop}$. After training a Transformer model with the language modeling objective, we test whether the model can generate the final answer without generating the intermediate steps. We experiment with different models of different sizes. From this, we can verify whether models of different sizes can acquire the *intuition* from the language modeling task.

We show examples of our data in Table 4. Each digit sequence represents a number from the lower digit to the higher digit. The reasoning process in the training set gradually updates both sides of "=" from the lowest digit to the highest digit. As for the testing example, we skip the intermediate steps, making the model complete right-hand side of "=" in the last step. The model needs to acquire *intuition* from the training set in order to solve the testing samples. (We include a rigorous description in Appendix C.)

Table 4: Training and testing examples for the digit addition task.

| Training | 4 9 2 8 4 6 + 0 8 0 3 5 0 = 0 0 0 0 0 0 ; 0 9 2 8 4 6 + 0 8 0 3 5 0 = 4 0 0 0 0 0 ; 0 0 2 8 4 6 + 0 0 0 3 5 0 = 4 7 1 0 0 0 ; 0 0 0 8 4 6 + 0 0 0 3 5 0 = 4 7 3 0 0 0 ; · · · 0 0 0 0 0 0 + 0 0 0 0 0 0 = 4 7 3 1 0 7 ; |
|---|---|
| Testing | 8 7 4 0 1 6 + 0 9 2 1 5 0 = 0 0 0 0 0 0 ; 0 0 0 0 0 0 + 0 0 0 0 0 0 = |

## 7.2 RESULTS AND DISCUSSION

We report the exact match accuracy and the digit-level accuracy of models trained with different $p_{drop}$ in Figure 3 with 5 random seeds. A higher accuracy implies the better intuition the model acquires from the sequence modeling task. The results show that 3 of the 4 models can acquire perfect intuition when $p_{drop}$ is as small as $0.3$. It suggests that it is possible that the model can acquire intuition from modeling step-by-step reasoning processes as in the case of modeling general text. We can also observe that larger models tend to have higher and more stable accuracy. Especially when the number of digits is 6 (Figure 10 in the appendix), only the largest model can acquire a perfect intuition. This observation is aligned with the emergence of large language models' ability.

## 8 RELATED WORK

Since Brown et al. (2020) discovered large language models' in-context learning ability, there have been some theoretical works attempting to explain how language models acquire this ability. Based on a hidden Markov model (HMM) assumption on the language generation process, Xie et al. (2022) suggested that in-context learning is an implicit Bayesian inference process. Hahn & Goyal (2023) defined the generation process with Compositional Attribute Grammar, which is weaker than the HMM assumption, explaining the in-context learning ability with the minimum description length. Zhang et al. (2023) assumed a more general latent variable model. Arora & Goyal (2023) analyze the emergence of skills based on the scaling law (Hoffmann et al., 2022). While their analysis assumes a set of skills as the atomic elements of NLP tasks, our hypothesis is based on a set of atom concepts.

There were also many empirical studies on the in-context learning ability. Some works focused on the effect of the instruction (Webson & Pavlick, 2022; Lampinen et al., 2022; Jang et al., 2023), while some focused on the examples in the demonstration (Liu et al., 2022; Lu et al., 2022; Sorensen et al., 2022; Min et al., 2022b; Yoo et al., 2022; Ye et al., 2023; Chang & Jia, 2023; Ye et al., 2023; Wang et al., 2023b). Shin et al. (2022) found that not all training corpora led to in-context learning ability. Prystawski & Goodman (2023) used synthetic data to suggest that the locality structure in the pretraining dataset contributes to the effectiveness of the reasoning steps. Wang et al. (2023a) studied the effect of reasoning steps in chain-of-thought.

Some people studied in-context learning as a meta-learning-like problem (Chen et al., 2022). Some works focused on the relationships between in-context learning and optimization algorithms (Garg et al., 2022; von Oswald et al., 2022). Akyürek et al. (2023) suggested that models perform in-context learning by implementing some optimization algorithms. Chan et al. (2022) studied the properties of dataset distribution that could contribute to the in-context learning ability. Li et al. (2023) provided generalization bounds based on the stability of Transformer models and the distance of downstream tasks. Compared to these works, we focus on how the pretraining data in natural language contributes to the in-context learning ability.

## 9 CONCLUSION

In this work, we propose a formalism for NLP classification tasks (§2). We also observe that modeling language involves the skill of recovering the underlying meaning of symbols based on a shared commonsense knowledge base (§3). We then propose a simple explanation for the in-context ability, the *Pelican Soup Hypothesis*, explaining in-context learning as the generalization of the language modeling task (§4). We provide supportive evidence in §5, §7 and §6. In addition to providing an explanation of the in-context learning ability, we believe our formalism in §2, observation in §3 and the datasets in §7 and §6 will help the future development of language models.

## REPRODUCIBILITY STATEMENT

For the generation process of Calcutec, please refer to § 5, Appendix A and Algorithm 1 in the appendix. For the hyper-parameters used for training, please refer to Appendix § B. The experiment in § 6 is based on the implementation of Min et al. (2022a). We will release the source code after the paper is accepted.

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

## A  PERTURBATIONS IN CALCUTEC

We apply two types of perturbations over the reasoning steps in Calcutec described in §5:

1. Random merge: At probability $p_{merge}$, for every two consecutive clauses where the consequence of the first one is in the antecedents of the second one, say $a_1 a_2 \rightarrow b_1$ and $b_1 a_3 \rightarrow b_2$, we merge them into a single clause $a_1 a_2 a_3 \rightarrow b_2$.

2. Random drop: Given a clause $a_1 a_2 \cdots a_n \rightarrow b$. We drop each of the antecedents $a \in \{a_1, a_2, \cdots a_n\}$ at probability $p_{drop}$. We apply this dropping to every clause in the proof except the first one to ensure that we do not drop the premises.

We use $p_{merge} = p_{drop} = p_{skip}$.

Additionally, when flatting the proof trees with topological sort, we break ties randomly. The order of the symbols in the antecedents is also random.

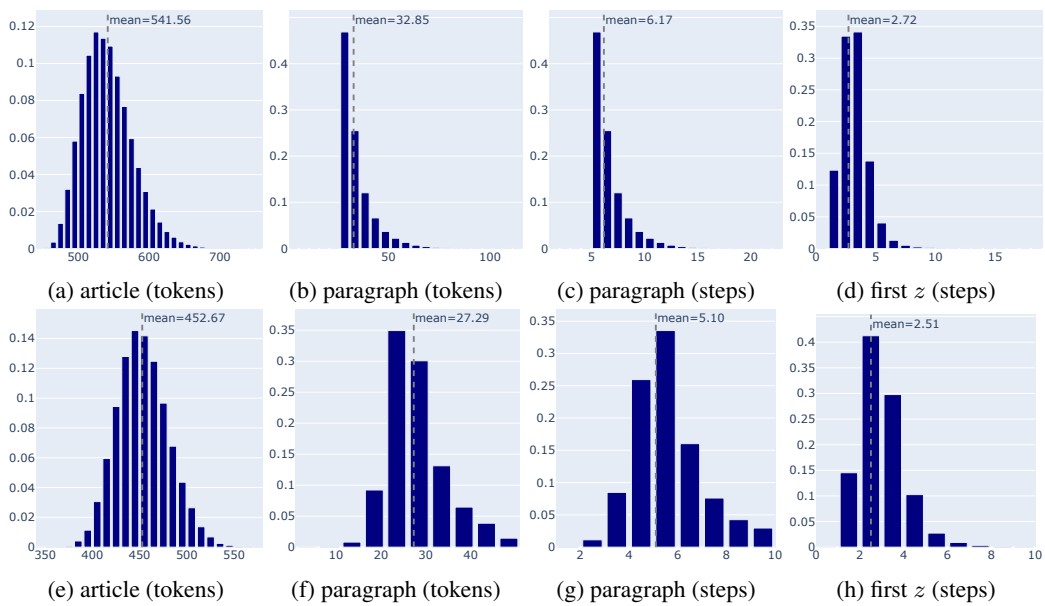

Figure 4: The distribution of lengths and the first step in each paragraph where $z$ is the consequence in the Calcutec dataset. The first/second row are the statistics before/after some steps are randomly dropped.

---

**Algorithm 1** Pseudo code for the generation process of an Calcutec article used for training.

---

Sample $r_a, r_b$ from $\{r_1, r_2, r_3, r_4\}$ with probability 0.45, 0.45, 0.05, 0.05.
Sample topic $S = \{s_1, s_2\} \subset \Sigma$.
Initialize a document $D$ with empty string.
**for** $p = 1, 2, \ldots, n_{par}$ **do**
  **while** True **do**
    Sample $s \in S$.
    Sample a set $X \subset \Sigma$ such that $\bigwedge_{x \in X} x \models s$.
    Run the resolution algorithm to get the set $M = \{m | X \models m\}$.
    Find an extra premise $x'$ that can increase the depth of deepest proof tree for $X \models m$.
    Run the resolution algorithm to get the set $M' = \{m | X \cup \{x'\} \models m\}$.
    **if** $|M'| > \frac{|\Sigma|}{2}$ **then**
      Reject the sampled $X \cup \{x'\}$.    ▷ *We don't want a premise that entails everything.*
      Restart the while loop.
    Sample a $g \in M'$ such that the proof tree for $X' \models g$ contains $s$ and its depth $> d_{min}$.
    ▷ *We use $d_{min} = 4$ in our experiments.*    ◁
    Do topological sort to flatten the proof tree and convert it into a string.
    Append the string to $D$.
**for** $s \in S$ **do**
  $D \leftarrow D.\text{replace}(s, r_a)$
Let $S' = \{s'_1, s'_2\} \in \Sigma$ be the top-2 frequent non $r_a$ symbols in $D$.
**for** $s' \in S'$ **do**
  $D \leftarrow D.\text{replace}(s', r_b)$

---

## B  HYPER-PARAMETERS

We train our model using batch size 256, warm up ratio 5%, and we truncate the sequence length to 512 tokens and the default parameters for the optimizer. We use the implementation of GPT-2 by Hugging Face transformers v4.27.2. All models can be trained with 4 RTX 2080ti within 8 hours.

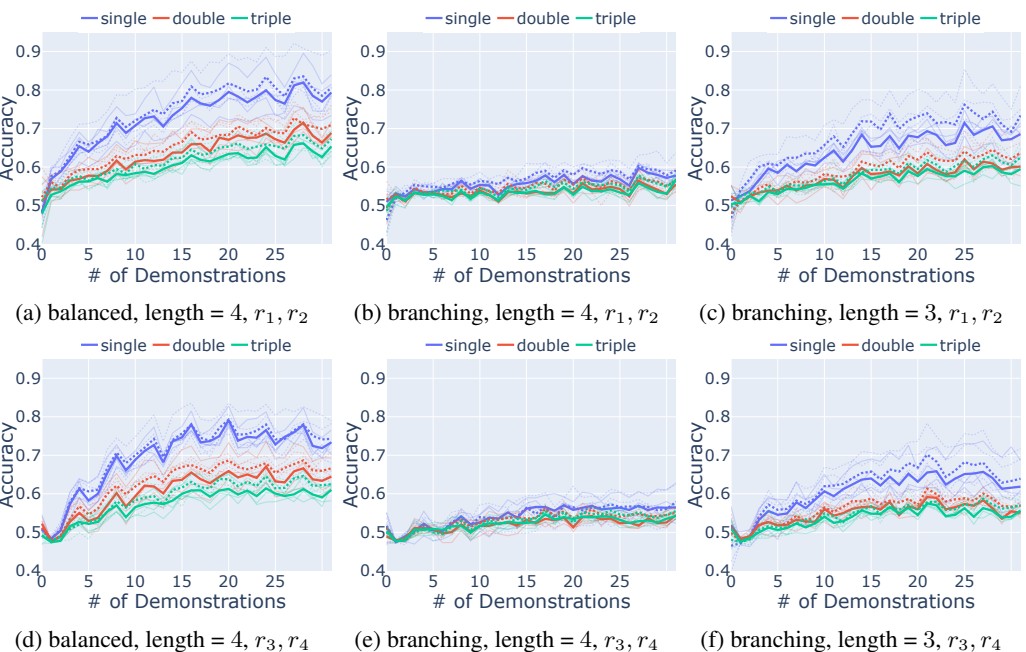

Figure 5: In-context learning accuracy with Calcutec when no steps are dropped ($p_{skip} = 0$). The dotted lines represent the performance of *unseen combinations* described in §5.3, while the different colors represent the number of atom concept in $\Sigma$ each verbalizer ($z_+$ or $z_-$) is associated to. The main lines represent the average accuracy of 5 tasks. We plot the performance of each task in lighter colors.

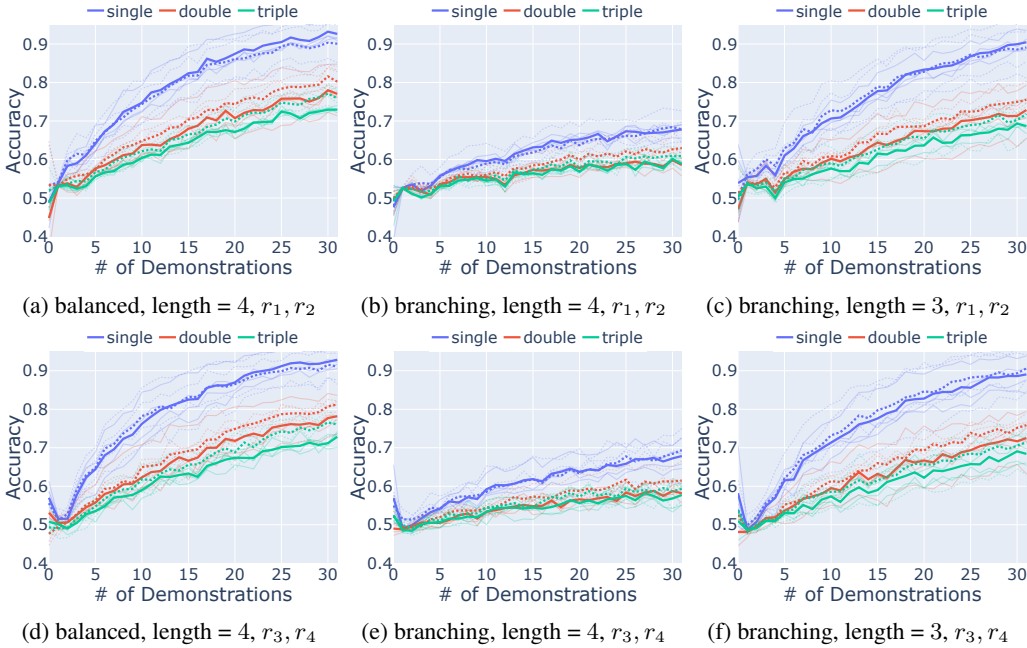

Figure 6: In-context learning accuracy with Calcutec without rewriting the first step to include contain the premise of the proof.

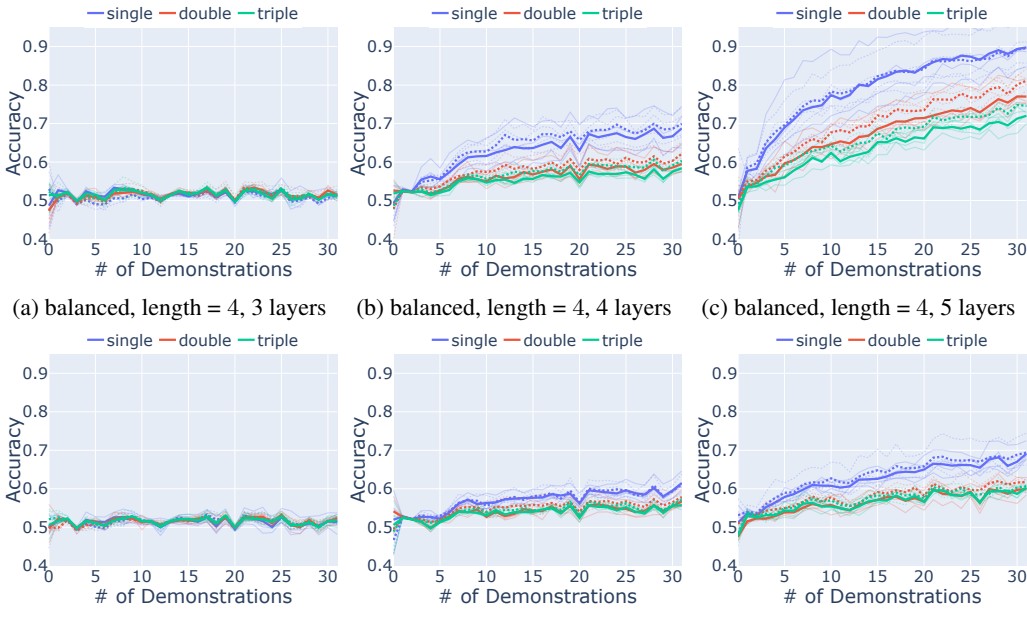

(a) balanced, length = 4, 3 layers  (b) balanced, length = 4, 4 layers  (c) balanced, length = 4, 5 layers

(d) branching, length = 4, 3 layers  (e) branching, length = 4, 4 layers  (f) branching, length = 4, 5 layers

Figure 7: The in-context learning performance when using models with different model depths.

Table 5: The 4-shot accuracy of in-context learning (ICL) versus chain-of-thoughts (CoT) when using $r_3, r_4$ as verbalizers.

| Task | Balanced | | Branching | |
|---|---|---|---|---|
| | ICL | CoT | ICL | CoT |
| single | 64.9 | 90.3 | 55.6 | 92.0 |
| double | 56.2 | 75.8 | 51.1 | 77.1 |
| triple | 54.2 | 67.0 | 51.7 | 73.4 |

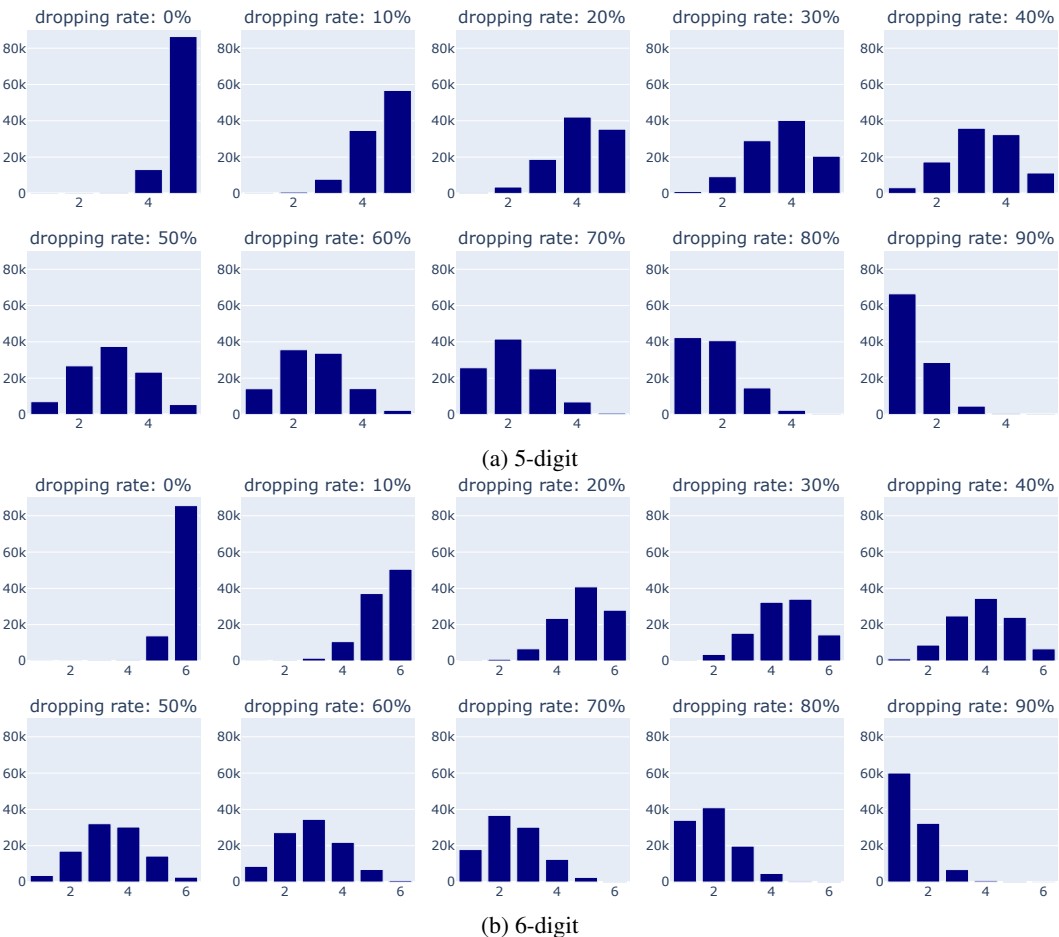

Figure 8: The distribution of the number of reasoning steps in the dataset when some of them are dropped at different probability. Each number is the average over 5 datasets generated with different random seeds.

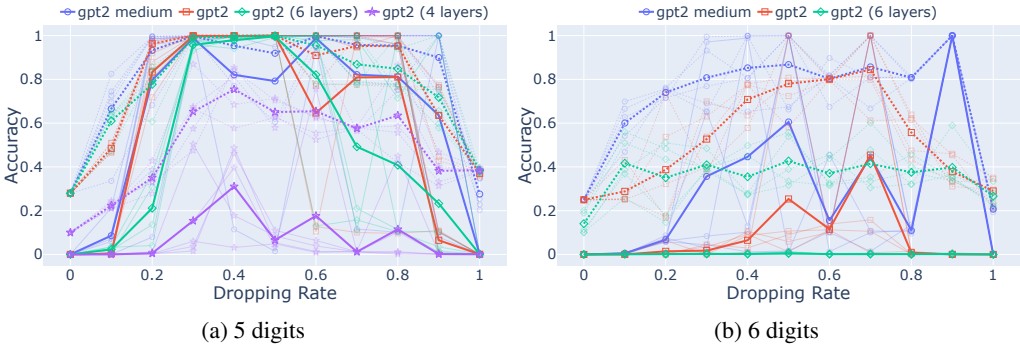

(a) 5 digits        (b) 6 digits

Figure 9: The accuracy of the models for the addition tasks. The x-axis represents the probability at which we drop each reasoning step in the training data independently. The solid line represents the ratio of testing samples where the model can output the exact answer, while the dashed line represents the character-level accuracy. The results are the average of 5 random seeds.

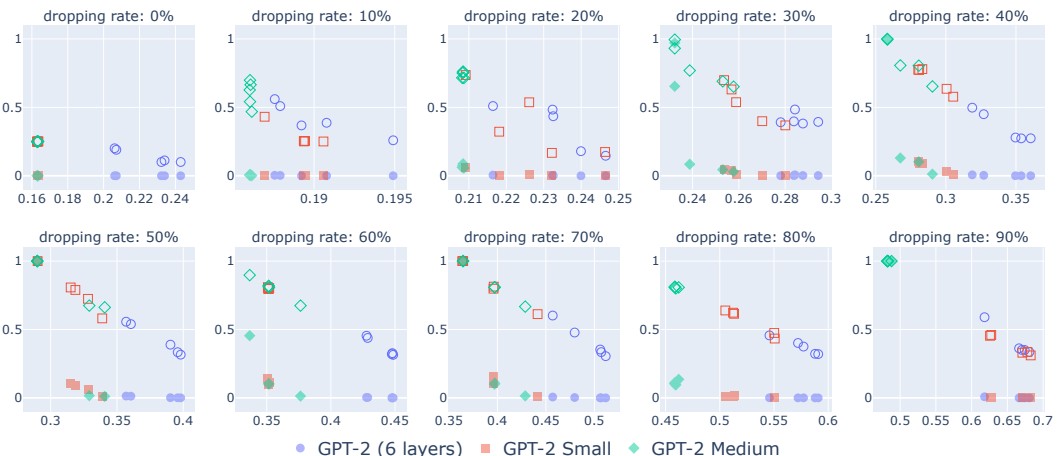

Figure 10: The exact accuracy (y-axis, solid points) and digit-level accuracy (y-axis, hollow points) versus validation loss (x-axis) for a 6-digit addition task.

## C FORMAL DESCRIPTION OF THE DIGIT ADDITION DATA

For each step $i$, we represent the step in the format $a^{(i)} + b^{(i)} = c^{(i)}$, where $a^{(i)}, b^{(i)}$ and $c^{(i)}$ are sequences of $n$ tokens, each of which is in $[0, 9]$, representing a number from the lowest digit to the highest digit. $a^{(0)}$ and $b^{(0)}$ represent two randomly drawn numbers and $c^{(0)}$ is all zero. At each step $i > 0$, most of the digit in $a^{(i)}, b^{(i)}, c^{(i)}$ is the same as the previous step. For $a^{(i)}$ and $b^{(i)}$, we only update the $i$th digit by setting $a_i^{(i)} = 0$ and $b_i^{(i)} = 0$. As for $c^{(i)}$, it serves as a buffer for both the answer and the carry. We update it based on $s^{(i)} = a_i^{(i-1)} + b_i^{(i-1)} + c_i^{(i-1)}$, the summation of the digits at $i$. We set $c_i^{(i)} = s^{(i)} \bmod 10$, $c_{i+1}^{(i)} = \lfloor s^{(i)}/10 \rfloor$. We use colons as the separator and concatenate these steps as a single sequence. When testing a model's intuition, we let the model generate the continuation for $a^{(0)} + b^{(0)} = c^{(0)}; a^{(n)} + b^{(n)} =$. Note that $a^{(n)} = b^{(n)} = 0$, so the model needs to have the *intuition* to generate the answer correctly. We provide examples in Table 4.

