# OpenReview forum: "Understanding In-context Learning with a Pelican Soup Hypothesis"
_ICLR.cc/2024/Conference — Submitted to ICLR 2024_

### Official Review · Reviewer_rNJR · 2023-10-27

**Soundness:** 3 good
**Presentation:** 2 fair
**Contribution:** 3 good
**Rating:** 6
**Confidence:** 2

**Summary:**

This paper focuses on understanding the in-context learning ability of large language models. The authors propose the Pelican soup hypothesis. It explains the in-context learning ability originating from learning the knowledge via the next token prediction. To support this hypothesis, the authors build a dataset and demonstrate the linkage between linguistic phenomena and in-context learning.

**Strengths:**

This paper provides substantial numerical results to support the proposed hypothesis. The linguistic phenomena analysis is also interesting to the community. In addition, the built dataset may be of independent interest.

**Weaknesses:**

1. The claim related to the knowledge base needs more clarification. The experiments in [1] demonstrate that input-output mapping is not very important to the ICL. If the label space is correct, LLMs can even implement efficient ICL given wrong mapping. However, this wrong mapping conflicts with the knowledge base. More discussions are needed here.

2. In Section 5.1, some assumptions are presented, but there is a notable absence of justification for these assumptions within the paper. This absence makes it challenging to ascertain the realism of these assumptions.
3. I would greatly appreciate further elucidation on the distinction between the hypothesis presented in this paper and that discussed in [2]. Specifically, the variance between the "atomic elements of NLP tasks" and "a set of atom concepts" requires additional clarification.

4. It is advantageous to include more highly relevant works in the related works. For example, besides HMM, implicit Bayesian inference is modeled for ICL in many different data assumptions [3,4,5]. [6] also studies the optimization side of ICL.

[1] Min S, Lyu X, Holtzman A, et al. Rethinking the role of demonstrations: What makes in-context learning work?[J]. arXiv preprint arXiv:2202.12837, 2022.

[2] Sanjeev A. and Anirudh G. A theory for emergence of complex skills in language models. arXiv preprint arXiv:2307.15936, 2023.

[3] Jiang H. A latent space theory for emergent abilities in large language models[J]. arXiv preprint arXiv:2304.09960, 2023.

[4] Zhang Y, Zhang F, Yang Z, et al. What and How does In-Context Learning Learn? Bayesian Model Averaging, Parameterization, and Generalization[J]. arXiv preprint arXiv:2305.19420, 2023.

[5] Wang X, Zhu W, Wang W Y. Large language models are implicitly topic models: Explaining and finding good demonstrations for in-context learning[J]. arXiv preprint arXiv:2301.11916, 2023.

[6] Dai D, Sun Y, Dong L, et al. Why can gpt learn in-context? language models secretly perform gradient descent as meta optimizers[J]. arXiv preprint arXiv:2212.10559, 2022.

**Questions:**

Questions are specified in Weakness part.

---

> ### Author Response · Authors · 2023-11-12
>
> Thank you for your comments.
>
> ## With Regard to the Weakness
>
> 1. We consider one main contribution of our work to be explaining the ICL ability in the *semantically-unrelated label ICL (SUL-ICL)* setting [8], which is a step forward from the theorem by [7]. Indeed, our work can not explain the random label setting in [1]. However, we would like to point out that as far as we know, there has been no theorem able to explain it. We suggest that explaining it requires making more assumptions on the relationship between the input-label mapping and the domain/topic/theme distribution in the training data.
> 2. We are sorry for the confusion. The justification mainly follows the argument in Section 3. We will make the connection clearer in our later revision. For now, please allow us to elaborate it below:
>     - Assumption 1: It follows the efforts made by early linguistics on linguistic formalism and cognitive psychology theories such as language of thought.
>     - Assumption 2: This assumption follows the argument in Section 3.1 that general text is usually organized in a similar way as logical induction processes. We assume that “a paragraph is a traversal of the nodes of the proving tree in the topological ordering” because in general text, statements usually follow some causal and/or chronological order, so it can convince readers or be easy for readers to understand.
>     - Assumption 3: In other words, this assumption assumes that every paragraph in an article mentions the topic of the article.
>     - Assumption 4: In other words, this assumption assumes that some pronouns are used in the article and each pronoun in the article is associated with a single entity.
>     - Assumption 5: This assumption follows the argument in Section 2.
>
>     In our opinion, these assumptions are relatively mild compared with previous theoretical works. We are happy to discuss more if you still have some concerns about the realism of these assumptions.
> 3. To make our main response short, please allow us to discuss it below.
> 4. Thank you for the list of relevant works. We will include some of them in our later revision (as we have some concerns about the technical soundness of some of them).
>
>
> ## Atomic elements of NLP tasks in [2]
>
>
> If we understand [2] correctly, [2] does not specify what characteristics the “atomic skills” of NLP need to have. We think our work could be an instantiation of the “atoms” proposed in [2].
>
> It seems that defining a set of “atomic skills” is not trivial and can’t be done arbitrarily. “Atomic skills” must be defined in a way such that the skills that work for the training data can generalize to the downstream task. To understand why it is non-trivial to define a set of atomic skills, we can probably think about a trivial case where we define the set “atomic skill” as the 26 skills to predict the words starting with A-Z. With this setting, the argument in [2] is greatly simplified. It seems that Theorem 14 only tells us that for most of the letters, the model can do the cloze problems in the training set well if the answer starts with that letter. It doesn’t seem to be guaranteed that the model can do the cloze problems in the testing set well.
>
> On the other hand, defining the atomic skills with our atomic concepts seems to be more generalizable. Still, we posit it is necessary to discuss the discrepancies between the training set and the downstream task. That’s why in this work we explicitly characterize the distribution shifts in Section 4 and demonstrate that LMs are able to generalize under these distribution shifts in Section 5 and 7. Surely our settings do not fully replicate the real-world data distribution, but we think it still complements some aspects of [2].
>
> Please let us know if you still have any questions. We are more than happy to discuss further.
>
>
> [7] Xie, Sang Michael, et al. "An explanation of in-context learning as implicit bayesian inference." arXiv preprint arXiv:2111.02080 (2021).
>
> [8] Wei, Jerry, et al. "Larger language models do in-context learning differently." arXiv preprint arXiv:2303.03846 (2023).

---

> > ### Comment · Reviewer_rNJR · 2023-11-23
> >
> > Thank the authors for the detailed response! It is encouraged to move the discussions of assumptions to the place near the assumptions. I will maintain my scores.

---

### Official Review · Reviewer_hbQa · 2023-10-29

**Soundness:** 3 good
**Presentation:** 3 good
**Contribution:** 3 good
**Rating:** 6
**Confidence:** 2

**Summary:**

This paper proposes the Pelican Soup Hypothesis to explain in-context learning. It says that the in-context learning in language models can be explained as generalisation under several types of distribution shifts. It provides a formalism of NLP classification tasks in the context of in-context learning and constructs a dataset in formal language demonstrating the hypothesis.

**Strengths:**

- It proposes a general formalism for NLP classification tasks in the context of in-context learning. As the paper says, it may facilitate future NLP theory research.
- The Pelican Soup hypothesis provides a potential explanation of in-context learning in language models.
- The Calcutec dataset may also facilitate future research on explaining in-context learning.

**Weaknesses:**

In general, I am not an expert in this line of work, but I have a strong feeling that the hypothesis and the experiment are more about mimicking, or more precisely, producing an environment, with which in-context learning still works, rather than explaining how/why in-context learning works in language models. Intuitively, for me, they are different things or at least an insufficient explanation.

Some of the reasonings are hard for me to follow, for example,
- why the yes/no questions are similar to the demonstrations for in-context learning? Or, such similarities had already been considered distribution shifts?
- (In page 3) How do you know the process of figuring out that "she" may be a person to whom something unexpected happened is similar to recovering z for class y? I understand the outcome would be similar, but why also the process?
- And if the above one is the actual process, I somehow feel that this suggests that LM should be good at handling anaphora but not catephora, which intuitively, is different from in-context learning.

**Questions:**

See my questions above.

---

> ### Author Response · Authors · 2023-11-13
>
> Thank you for your positive review.
>
> We would like to point out that producing an environment, with which in-context learning still works, is a practice used in most (if not all) theoretical analyses such as [1] and [2] and even some empirical analyses such as [3]. We consider one of our main contributions to be proposing an environment that is closer to the real-world scenario than previous works.
>
> With regard to the questions you stated:
>
> - The yes/no questions in a Pelican Soup riddle have a similar function as the demonstrations for in-context learning, because based on the answer of those yes/no questions, the participants are able to figure out the latent story and thus are able to answer other following questions. Note that we use Pelican Soup riddles just to motivate the importance of the role of commonsense.
> - Our argument is that solving these two problems (ICL and modeling coreference) requires similar capabilities. This argument is to draw the intuition for the framework/assumptions in Section 5.1. Surely the two problems are not identical, thus we characterize the difference between these two problems with the distribution shifts discussed in Section 4. To show that LMs may be able to generalize under these distribution shifts, we replicate these distribution shifts in our experiments, as discussed in Section 5.3 and results in Section 5.5 are aligned with our hypothesis.
> - We understand ICL as generating verbalizers based on the meaning of the verbalizers, which is recovered according to the preceding context, so we think it is similar to modeling anaphora. Could you please elaborate more on why you think ICL is more like handling catephora?
>
> Please let us know if you have any other comments. We are happy to discuss more!
>
> [1] Xie, Sang Michael, et al. "An Explanation of In-context Learning as Implicit Bayesian Inference." International Conference on Learning Representations. 2021.
>
> [2] Hahn, Michael, and Navin Goyal. "A theory of emergent in-context learning as implicit structure induction." arXiv preprint arXiv:2303.07971 (2023).
>
> [3] Chan, Stephanie, et al. "Data distributional properties drive emergent in-context learning in transformers." Advances in Neural Information Processing Systems 35 (2022): 18878-18891.

---

> > ### Comment · Reviewer_hbQa · 2023-11-16
> > **Response to Rebuttal**
> >
> > Thanks for your response.
> >
> > > We would like to point out that producing an environment, with which in-context learning still works, is a practice used in most (if not all) theoretical analyses such as [1] and [2] and even some empirical analyses such as [3]. We consider one of our main contributions to be proposing an environment that is closer to the real-world scenario than previous works.
> >
> > Thanks! This gives me a clearer picture of how this line of studies works.
> >
> > > Surely the two problems are not identical, thus we characterize the difference between these two problems with the distribution shifts discussed in Section 4. To show that LMs may be able to generalize under these distribution shifts, we replicate these distribution shifts in our experiments, as discussed in Section 5.3 and results in Section 5.5 are aligned with our hypothesis.
> >
> > I still feel that an explanation about this approximation or assumption is missing here, which should be seen as a limitation later on.
> >
> > > ICL is more like handling cataphora?
> >
> > btw. I didn't say ICL is more like cataphora. Maybe you misread my review. (Maybe because I misspelled cataphora as catephora. My apologise!)

---

### Official Review · Reviewer_5jds · 2023-11-01

**Soundness:** 2 fair
**Presentation:** 2 fair
**Contribution:** 2 fair
**Rating:** 3
**Confidence:** 4

**Summary:**

This paper introduces the Pelican Soup Hypothesis to formalize and explain large language models' ability for in-context learning. The paper claims that in-context learning can be seen as a model's ability to generalize linguistic phenomena under distribution shifts. The authors identify several contributions in the paper, including:
1. A new formalism for approaching natural language classification problems, specifically aimed at understanding in-context learning.
2. A new dataset, "Calcutec," replicates specific linguistic phenomena. The authors report that training on this dataset allows models to develop in-context learning abilities and improve their performance on chain-of-thought reasoning.
3. The paper reports experiments with the GPT-2 model on various NLP tasks. These experiments connected certain linguistic phenomena and the model's in-context learning capabilities.
4. The authors use a digit addition task to study a specific type of distribution shift. This experiment revealed that larger models are more capable of generalizing and adapting to such shifts.

**Strengths:**

1. Linking in-context learning to the model's coreference learning ability is an interesting and novel idea. It serves the vast interest of the community in understanding the underlying mechanism of LLMs' in-context learning and chain-of-thought ability.
2. Overall, the Pelican Soup Hypothesis and the accompanying experiments provide insights into why and how in-context learning works in large language models. The introduction of the Calcutec dataset and the digit addition task as experimental tools paves the way for further research in this area.

**Weaknesses:**

1. Many claims are made without citing prior sources or supporting evidence. For example, in 3.1, the author claims that “language models may be able to acquire the KB by modeling general text.” However, no clear evidence is provided via citations or experiments, and frankly, this is still an ongoing question the community aims to answer; it would be an important work itself to show these claims.

2. The pretraining and in-context learning setting in the proposed dataset is different from common LLM settings, in which the synthetic setting here loses some of the information that LLM encodes, such as contextual information and domain information. This mismatched setting seems not ideal and limits the generalizability of this study. In particular, in-context learning has been found to be highly sensitive to context-label and domain-label biases, which is not clear in a context-free & domain-free setting.

3. The main assumption of this paper seems to be that the text in pretraining corpora for LLMs consists of clear reasoning steps (potentially with some intermediate steps dropped). However, this assumption normally requires structured and domain-specific training data such as math text or academic papers. On the other hand, data like dialogues or other internet content may contain completely implicit reasoning steps that are hidden in the text space. So, I don't think the proposed pretraining data here, which includes some reasoning steps explicitly in the sequence, is very representative of the overall LLM pretraining setting.

4.  The experiments are poorly designed, and the implementation details are generally missing, but the main experiment on Calcutec: dataset design is too complicated, but the experimental design and analysis are too simple, although the fact that it can do in-context learning is interesting. In addition, what is discussed in section 6 as real-world evidence does not directly support their main hypothesis.

5. The logic of the paper is weak, and the paper is poorly organized. The arguments are not supported by rigorous experimental evidence. Almost all arguments around using the word "**Therefore**" are not rigorous (either the conclusion is not supported by the evidence or the things after, therefore, are logically irrelevant to things before). A large number of arguments are based on the author's thinking that A is **similar** to B, where first the similarity is poorly defined, and how from such similarity can we conclude their conclusion is usually unclear. For, in section 3.2, the author claims that predicting the correct pronoun in the next token completion using the information in the context is "**similar**" to inferring the class description z_y for y in text classification. "**Therefore**" modeling general text is similar to performing in-context learning. This may "explain" the linkage between in-context learning and emergent abilities of LLMs.

6. The title of the work or the main motivation: human solving Pelican Soup riddles is similar to LLM doing in-context learning is based on some poorly defined subjective similarity.

7. Could design more controlled experiments to study the importance of each individual aspect of the dataset (the current construction of the dataset is too complicated) and also to rule out other possibilities. For instance, the binary classification problem seems a bit too easy. Can the model learn shortcuts instead of using their "world" knowledge to solve the problem?

**Questions:**

1. What's your view on the mesa-optimization view of in-context learning based on your Pelican Soup Hypothesis? Do they complement each other, and can one explain the other one?

---

> ### Author Response · Authors · 2023-11-13
>
> We thank you for your review. We would like to address the weakness below:
>
> 1. Please let us know what are the arguments that you think are not substantiated. For the example you provided, we apologize for not including proper citations. We think that LMs are able to acquire some knowledge about commonsense (though imperfectly) has been widely accepted. Many studies can be found in previous works, e.g. [1][2].
> 2. Validating theories with a synthetic dataset is a common practice. We admit that our setting cannot explain domain bias. It may require more assumptions to theoretically explain. We would like to mention that, we think it will make the construction of the toy data even more complicated, which based on your 4th point, is not acceptable.
> 3. We justify our assumptions made in Section 5.1 in Section 3. As argued in Section 3.1, we posit that training data usually contain text that has some logical structure and/or follows chronicle order, which can contribute to LMs’ reasoning capability. The dataset Calcutec is a formal abstraction of such structure.
> 4. Could you be more specific on what details are missing and how we should improve? We will appreciate your suggestion. The construction of Calcutec is more complicated than the toy data used in previous theoretical works such as [3] because Calcutec relies on milder assumptions and is more realistic.
> 5. We apologize that we didn’t make the purpose of Section 3 clearer. We treat Section 3 as the justification/intuition for the abstraction/assumptions/framework we made in Section 5.1. We then instantiate the abstraction/framework with Calcutec and show that LMs trained with it can do in-context learning.
> 6. We use Pelican Soup riddles only to motivate the intuition that commonsense plays an important role for ICL. We would not say our hypothesis is *based on* this similarity.
> 7. Please let us know if you think any experiment is important but missing. Also, could you elaborate more on the possibility that the models solve the task by learning shortcuts (and maybe also your definition of shortcuts)?
>
> Our answer to the question
>
> 1. We think mesa-optimization mainly explains why models of Transformer architecture trained with autoregressive loss have the capacity to model data that involves ICL-like phenomena. However, in works such as [4], they only study on some toy data. It is unclear how their toy data is relevant to natural language data. Our work starts from another direction, aiming to understand what characteristics of training data lead to the ICL ability.
>
>
> [1] West, Peter, et al. "Symbolic knowledge distillation: from general language models to commonsense models." arXiv preprint arXiv:2110.07178 (2021).
>
> [2] Li, Xiang Lorraine, et al. "A systematic investigation of commonsense knowledge in large language models." Proceedings of the 2022 Conference on Empirical Methods in Natural Language Processing. 2022.
>
> [3] Xie, Sang Michael, et al. "An explanation of in-context learning as implicit bayesian inference." arXiv preprint arXiv:2111.02080 (2021).
>
> [4] von Oswald, Johannes, et al. "Uncovering mesa-optimization algorithms in transformers." arXiv preprint arXiv:2309.05858 (2023).

---

> > ### Comment · Reviewer_5jds · 2023-11-23
> >
> > Thank you for the detailed response!
> > > More statements that need evidence and citation
> > 1. 'language models may learn to do reasoning with the rules in the KB." Yes, some models do mimic this kind of rule-based reasoning from purely pretraining, but a lot of models fail to show a robust ability for rule-based reasoning. This statement doesn't seem to generalize well enough to support the border scope of the paper's main motivation, i.e., explain in-context learning.
> > 2. "Therefore, by modeling these articles, a language model can not only learn the commonsense rules in KB but also
> > learn to utilize these rules for induction." How can you be sure that the model learns the commonsense rules and the ability to reason based on these rules purely from the articles? There are other possibilities as well, such as code data and textbooks, etc. How is the model's ability to "utilize these rules for induction" being evaluated?
> > 3. "Such kind of articles may be pervasive in the training data. Essays arguing some claims are one
> > example." How can you be sure that articles in the pretraining data are pervasive? Most of the models with strong in-context learning ability do not open source their pretraining data.
> >
> > > The construction, assumptions, and motivations of Calcutec
> > I agree that the construction of this dataset is complex, but I fail to see that this dataset is more realistic and the relevance between building a formal abstraction of the logical structure and the pretraining data that are used in current work for Large Language Model pretraining.  I think you would need to show that (1) such logical structure largely exists in the pretraining data of current LLMs and (2) the data of such logical structure show certain levels of impact on the model's behavior post-pretraining.
> >
> > > Models solve the task by learning shortcuts
> > Mainly domain-label bias, context-label bias, and vanilla-label bias. See https://arxiv.org/abs/2305.19148 and https://arxiv.org/abs/2102.09690.
> >
> > Overall, I will raise my soundness to 2 while keeping my original rating.

---

### Official Review · Reviewer_MUGa · 2023-11-01

**Soundness:** 3 good
**Presentation:** 3 good
**Contribution:** 2 fair
**Rating:** 5
**Confidence:** 4

**Summary:**

The paper presents a new theoretical account of the in-context learning (ICL) abilities of large language models.
Section 2 describes a formal framework for NLP classification tasks, inspired by commonsense knowledge bases.
Section 3 intuitively discusses, using this framework how the structure of language may lead to ICL abilities.
Section 4 specifically describes three ways in which ICL shows a distribution mismatch relative to general language modeling.
Sections 5--7 adduce experimental evidence from three domains: a new synthetic dataset ("Calcutec"), evidence from a small LMM, and a digit addition task.

**Strengths:**

- I found the toy dataset ("Calcutec") quite interesting, and to improve in some ways over prior synthetic setups for ICL, such as Xie et al 2022 or Chan et al 2022, in that it includes a simple kind of logical reasoning.

- Provides evidence that even smaller LLMs (GPT-2) can perform ICL with artificial/task-agnostic label symbols (which Wei et al 2023 argued only large LLMs can do).

- provides empirical results from different domains

**Weaknesses:**

- While I found the Calcutec experiment in particular to be innovative, the theoretical arguments in Sections 2--3 are quite hand-wavy and unspecific. There is no rigorous theoretical statement of the assumptions and conclusions made in the theoretical framework and the reasoning of how language modeling may lead to ICL.

- While I believe the Calcutec toy dataset is an interesting contribution and a strength of the paper, it is limited in that the training dataset appears to bake in the repetitive nature of prompts by assuming that each "paragraph" in a document is about one of two latent concepts ("topics"), as in the prompting downstream tasks. A potential concern about the CoT evaluation is mentioned as a Question.

- In the Digit Addition Task, the ability of the LM to complete the task in one go, whereas the training set usually had intermediate steps, is interpreted as an ICL ability representing a domain shift. However, as the training set also had intermediate steps stochastically dropped (independently, as far as I got from the paper -- so it is possible for all steps to be dropped simultaneously), it is not clear in which sense the test examples are out-of-domain relative to the training distribution. The same concern applies to the Calcutec dataset.

**Questions:**

- How exactly is Chain-of-thought evaluated in Calcutec? Does the prompt only include the first step in the chain? And under what circumstances is the LM's answer counted as correct -- are predictions rolled out until the ";" paragraph appears? This question is crucial for assessing the meaningfulness of the CoT results.

---

> ### Author Response · Authors · 2023-11-12
>
> Thank you for your thoughtful comments. We would like to address the weakness below:
>
> 1. In Section 2, we propose a formalism for NLP classification tasks while the main purpose of Section 3 is to justify the assumptions made in Section 5. We see the assumptions in Section 5 as the formal statement/abstraction of our theorem in Section 3. We are sorry that we didn’t describe it more clearly. We will improve this part in our later revision. Please let us know if we need to make more clarifications.
> 2. As far as we know, all theoretical analyses on ICL depend on some repetitive nature of prompts or training data, e.g. Xie et al and Hahn & Goyal. One difference in our work is that we focus on the repetitive usage and the consistent meaning of pronouns, which is very realistic in real-world data. In Figure 2, we also show that the model can still do ICL when the task description is more complicated than the meaning of the pronouns seen in the training set, i.e., when the task description is composed of 3 atomic concepts (the green lines for “triple” in Figure 2).
> 3. We discuss this point below:
>     - The digit addition task: Please refer to Figure 8 for the distribution of the number of steps in the datasets. For the 5-digit addition task, when the dropping rate is 0.2, there are only 265 (0.265%) training samples where all the intermediate steps are dropped. We think it is reasonable to say the testing samples are out-of-domain in this case. (And GPT-2-sized model can achieve ~80% accuracy with only 265 training samples in the testing domain.) Meanwhile, considering that the real-world data also has a small fraction of text that drops all the reasoning steps, we think this setting is not unrealistic.
>     - Calcutec: Please refer to Figure 5. It shows that even when no steps are dropped in the training set, the model still can do ICL, though the performance is not as good.
>
> Question:
>
> - The prompt is as the one in Table 1. We only include the reasoning steps of some (3) randomly selected examples and the premise of the testing input in the prompt. (For the example in Table 1, the testing input is `x55 x76 x84 x99`). We unroll until the first verbalizer is generated.
>
> Please let us know if you have any other questions. We are more than happy to discuss with you.

---

> > ### Comment · Reviewer_MUGa · 2023-11-18
> >
> > Thank you for the response.
> >
> > The Question is answered to my satisfaction.
> >
> > Regarding the Weaknesses, I take your point that other theoretical studies of ICL also need to assume repetition in the training set. I also appreciate the explanations regarding the number of steps. In response to this, I have increased the 'soundness' score to 3.

---

### Official Review · Reviewer_cdtQ · 2023-11-02

**Soundness:** 3 good
**Presentation:** 3 good
**Contribution:** 3 good
**Rating:** 6
**Confidence:** 4

**Summary:**

The paper proposes the "Pelican Soup Hypothesis" to explain in-context learning in large language models. The key idea is that in-context learning relies on models acquiring commonsense knowledge and reasoning skills from pretraining on general text. The paper formalizes NLP classification tasks as mapping inputs to output concepts based on commonsense rules and knowledge. Experiments on a synthetic dataset Calcutec show models can acquire in-context learning abilities.

**Strengths:**

1. This paper provides a clear and intuitive conceptual framework based on the Pelican Soup analogy to explain in-context learning.
2. The proposed formalism for NLP tasks is simple yet quite general. It could be a useful tool for future theory research.
3. Evidence from synthetic data, language modeling, and a toy task provide empirical support for the central hypothesis.
4. The Calcutec dataset offers a nice testbed for studying in-context learning and model architectures.
5. Analysis of the digit addition task sheds light on how model scale impacts reasoning abilities.

**Weaknesses:**

1. The explanations are conceptual. More formal theoretical analysis could better elucidate the mechanisms.
2. More analysis could be done on how different pretraining corpora impact in-context abilities.
3. The hypothesis focuses on classification; generative tasks may involve additional factors.

**Questions:**

1. Can we quantify the relative importance of different distribution shifts identified?
2. How well does the formalism proposed capture more complex real-world reasoning?
3. Is it possible to design pretraining objectives to better acquire commonsense and reasoning?
4. How can we test if models learn explicit commonsense rules and reasoning versus pattern matching?

---

> ### Author Response · Authors · 2023-11-13
>
> We thank you for the positive feedback. With regard to the weakness:
>
> 1. While Section 3 is purely conceptual, we provide rigorous characterization in Section 5.1. We consider this as one of our main contributions, an instantiation for the assumptions made in previous theoretic works, such as [1] and [2]. For [1], our framework constituted with the assumptions we made in Section 5.1 satisfies the requirements of Corollary 4.2, therefore we can use their analysis to get an $O(1/T)$ regret bound ($T$ is the number of examples in the demonstration). For [2], please refer to our reply to reviewer rNJR.
> 2. Thanks for the suggestion! We can run more experiments. Could you let us know what experiments you think are important are missing?
> 3. We admit that we only focus on classification tasks in this work. However, we would like to stress that even for the classification setting, the mechanism of ICL is not yet clear. Future work may also extend our framework to generation tasks.
>
> With regard to the questions:
>
> 1. By preprocessing the training data for LLMs, we think it’s possible. However, training an LLM requires lots of computational resources. We leave it for future work.
> 2. The formalism in Section 2 can handle arbitrarily difficult reasoning tasks as long as there is a logic system that can handle it and the induction searching process is computationally feasible, e.g. not NP-hard. We believe that most human-solvable tasks are in this class.
> 3. We think this is an important open question. A possible solution would be augmenting the training data, e.g. [3].
> 4. Our conjecture would be that for many commonsense rules, LLMs are trained with many instances for each of them. Because the training data may cover a large portion of common surface forms, even though it’s possible that LLMs are doing no more than pattern matching, they are still able to do “reasoning” following the commonsense rules.
>
> [1] Zhang Y, Zhang F, Yang Z, et al. What and How does In-Context Learning Learn? Bayesian Model Averaging, Parameterization, and Generalization[J]. arXiv preprint arXiv:2305.19420, 2023.
>
> [2] Sanjeev A. and Anirudh G. A theory for emergence of complex skills in language models. arXiv preprint arXiv:2307.15936, 2023.
>
> [3] Zhou, Wangchunshu, Ronan Le Bras, and Yejin Choi. "Commonsense Knowledge Transfer for Pre-trained Language Models." arXiv preprint arXiv:2306.02388 (2023).

---

### Author Response · Authors · 2023-11-21
**Overall response**

We appreciate all reviewers' insightful comments.

We are particularly grateful that the reviewers agreed that our paper has the following strengths:

- The linguistic phenomena analysis is interesting to the community (rNJR, MUGa, 5jds).
- The Calcutec dataset and the digit-addition task may be useful for future research (rNJR, MUGa, 5jds, hbQa,cdtQ).
- Provides evidence that even smaller LLMs can perform ICL with artificial/task-agnostic label symbols (MUGa, 5jds).
- A new formalism for NLP classification tasks (5jds, hbQa, cdtQ).
- Empirical results supporting the Pelican Soup Hypothesis (rNJR, cdtQ) and provide a potential explanation for in-context learning (hbQa).

We have revised the paper to address the main concern that Section 3 is not substantiated, which we acknowledge is the confusion caused by the inclarity. We added more elaboration on the linkage between Section 3 and Section 5.1, stressing that Section 3 is to draw intuitions for the assumptions in Section 5.1 and that Section 5.1 is the formal description for the characteristics described in Section 3.

We also address each reviewer’s concerns below respectively. Please let us know if any aspects are still unclear. We are more than happy to provide additional clarification.

---

### Meta-Review · Area_Chair_9xsW · 2023-12-08

**Metareview:**

This paper introduces the Pelican Soup Hypothesis to formalize and describe to in-context learning abilities of LLMs. Specifically, the paper argues that in-context learning is a LLM's ability to generalize linguistic phenomena under distribution shifts. In addition to this formalization of ICL, the authors propose a new dataset, Calcutec, which replicates specific linguistic phenomena and allows models to develop in-context learning abilities when trained on it.

The reviewers generally found the attempt to conceptualize ICL interesting, and appreciated the Calcutec dataset and the empirical results. However, there were recurring concerns among reviewers that the connections between the experiments and the theory in Sections 2 and 3 was unclear and hand-wavy. The authors have made improvements to this section during the rebuttal period, but the reviewers that checked this update were not swayed by the changes.

**Justification For Why Not Higher Score:**

The connections between the conceptual argument in Sections 2 and 3 and the experiments in Section 5 could be stronger. As such, while the reviewers agreed that the paper had some strong contributions with the Calcutec dataset and empirical results, they felt the overall framing needed to be re-worked to clarify the conceptual grounding.

**Justification For Why Not Lower Score:**

N/A

---

### Decision · Program_Chairs · 2024-01-16

Reject